# Porosity Analysis of Additive Manufactured Parts Using CAQ Technology

**DOI:** 10.3390/ma14051142

**Published:** 2021-02-28

**Authors:** Peter Pokorný, Štefan Václav, Jana Petru, Michaela Kritikos

**Affiliations:** 1Faculty of Materials Science and Technology in Trnava, Institute of Production Technologies, Slovak University of Technology in Bratislava, 917 24 Trnava, Slovakia; stefan.vaclav@stuba.sk (Š.V.); michaela.kritikos@stuba.sk (M.K.); 2Faculty of Mechanical Engineering, VŠB—Technical University of Ostrava, 70800 Ostrava, Czech Republic; jana.petru@vsb.cz

**Keywords:** porosity, additive technology, SLM, computer tomography

## Abstract

Components produced by additive technology are implemented in various spheres of industry, such as automotive or aerospace. This manufacturing process can lead to making highly optimized parts. There is not enough information about the quality of the parts produced by additive technologies, especially those made from metal powder. The research in this article deals with the porosity of components produced by additive technologies. The components used for the research were manufactured by the selective laser melting (SLM) method. The shape of these components is the same as the shape used for the tensile test. The investigated parts were printed with orientation in two directions, Z and XZ with respect to the machine platform. The printing strategy was “stripe”. The material used for printing of the parts was SS 316L-0407. The printing parameters were laser power of 200 W, scanning speed of 650 mm/s, and the thickness of the layer was 50 µm. A non-destructive method was used for the components’ porosity evaluation. The scanning was performed by CT machine METROTOM 1500. The radiation parameters used for getting 3D scans were voltage 180 kV, current 900 µA, detector resolution 1024 × 1024 px, voxel size 119.43 µm, number of projections 1050, and integration time 2000 ms. This entire measurement process responds to the computer aided quality (CAQ) technology. VG studio MAX 3.0 software was used to evaluate the obtained data. The porosity of the parts with Z and XZ orientation was also evaluated for parts’ thicknesses of 1, 2, and 3 mm, respectively. It has been proven by this experimental investigation that the printing direction of the part in the additive manufacturing process under question affects its porosity.

## 1. Introduction

Extensive funds are invested in additive technologies. Large corporations are building laboratories focused on research and the application of additive technologies. For example, it is possible to cite the Centre for Additive Technology [1]. Components made with additive technologies are also implemented in the automotive and aerospace industries; for example, the use of additive manufacturing to manufacture fuel nozzles [2]. Additive technologies for the production of metal components use various metal powders. Bajaj et al. dealt with steels used in the additive manufacturing process in their experimental investigation. The article compares some mechanical properties of components produced by additive technology and conventional technology. It was proven that some properties (hardness, corrosion resistance) are better for components made with additive technologies. Some properties, such as ductility or fatigue strength, are worse for components made by additive technologies in comparison to conventionally produced steel parts [3]. One of the kinds of steel powders used for additive production is 316L. Similarly, other authors have researched and described the change in the mechanical properties and change of the structure of parts produced by the selective laser melting (SLM) method [4]. Through the analysis of mechanical properties, these authors were able to determine the direction in which these properties were changing. Additionally, anisotropic properties of steel were studied in different papers [5]. These papers specifically examined stainless steel. The authors investigated the properties of components produced by additive technology that were oriented at different angles during production. They found out that components made at an angle of 45° had the highest tensile strength. Research has shown that the angle of orientation of a part during its production and the direction of its fibres are influenced by the tensile characteristics. In a similar way, the heat treatment, mechanical, and microstructural properties of stainless steel were discussed in investigations by the authors in [6,7,8]. On the other hand, the parameters of the production process in the production of components by additive technology are important. These include, for example, distance of points and time of exposure. The combination of these parameters has also an influenced the porosity, component surface, microstructure of the material, density, and hardness. The parameters of the sintering process of the powders were investigated [9]. These authors found that, with the increase of the laser energy, the temperature at the sintering site increases and thus the melt fills the voids, leading to a porosity reduction in the component. Another important parameter that affects the properties of the component is the scanning strategy used in the producing process, which in this case, is basically a computer aided manufacturing (CAM) strategy. CAM strategies are also used in conventional machining methods; for example, in the milling process. One article [10] shows the influence of milling strategies on surface accuracy. In this work, a part with simple shapes was modelled. Three finishing strategies (optimized constant Z, spiral finishing, and offset finishing) were applied to these shapes. Afterwards, the measuring of machined cylindrical surfaces was performed by the optical 3D scanner and then by Contura G2. It is clear from the results that the optimized constant Z milling strategy is the most suitable for a simple cylindrical surface. The smallest deviations were recorded in both types of measurements for this strategy. It is actually a movement along the shape of the part, which removes the material during machining and adds material during scanning in additive production. Similarly, authors [11,12] investigated the influence of scanning strategy parameters on residual stress by SLM technology and the impact of process on the final component properties. Three scanning strategies were used (chessboard, stripes, and the meander strategy). A high density of sintered material of up to 99.695% was found. The effect of the laser power on the residual stress was also proven. In terms of the grain structure, this was observed and detailed in the article [13], where the influence of two scanning strategies on the grain structure in the material was studied. The authors used two scanning strategies (island and back and forth). The study showed that a more homogeneous structure can only be achieved by changing the scanning strategy. Here, scanning strategies were applied to a simple sample shape (block). The structure may develop differently on a sample with more complex shape. Mechanical properties of AISI 316 stainless steel engaging different orientation of parts were presented in [14]. In this case, the samples were made with different orientations with respect to the machine platform. The sintering parameters were constant for all samples, while the orientation was the only parameter changing. Measurements have shown that some mechanical properties (such as strength) of steel produced by additive technology are better than those of steel produced by rolling. Similarly, these samples showed anisotropy of properties with respect to their orientation during production. Also, an important property of components is their porosity. The article [15] investigates porosity and microhardness of 316L. When examining the porosity, the areas on the samples with defects were evaluated. One sample was examined by the non-destructive X-ray computer tomography (XCT) method. The other samples were subjected to metallographic examination. A high sample density of more than 99% and a low porosity of about 0.82% were investigated. The pores were not evenly distributed. The samples had higher microhardness than the parts made by molding. Still, in this regard, the authors in [16] investigated density and porosity of sintered steel. The article analyzed particle size, particle shape, temperature of sintering, and time of sintering. These parameters affect the properties of the parts. They determined the relationships between the parameters of the sintering process and the properties of the parts. The authors [17] described the possibility of using computer tomography (CT) to evaluate the properties of parts produced by additive technology. They analyzed the use of CT from several perspectives (defects, dimensions, density, and roughness). Based on an extensive analysis of the use of CT to measure the properties of parts produced by additive technology, they made suggestions for evaluating the relationship between the production of parts and their mechanical behaviour. In addition, they concluded that international standards for the use of CT for printing technology needed to be set; that it was necessary to specify a reference element for CT calibration; and that due to the high cost of CT machines, it was necessary to consider using other cheaper measurement methods as well to establish standards for examining the surface of parts via CT; and, thus, developing a software tool to simulate the CT process would be of great aid. In the article [18], the authors used CT to investigate the properties of parts. Samples were made by PμLSE stereolithography. They used CT to examine surface defects and lattice defects. Thus, they predicted the behaviour of mechanical properties and defects in the lattice. They performed shearing experiments. They summarized the findings of the experimental investigation in several points, concluding that geometry defects were related to the direction of the part building, and that geometry defects affected the development of defects in the material grid itself. On the other hand, the results of their finite element simulations showed the influence of geometry on the formation of defects in the material lattice.

In this paper, we focused on the research of porosity in components made by additive technology, specifically, SLM technology. The motivation for this research was the study of materials and scientific publications on the porosity of materials produced by SLM. These studies have shown that porosity needs to be deeper examined given that it can significantly affect the mechanical properties of parts. The study of porosity and other properties of parts made by 3D printing must lead to quality products that will be produced in a shorter time.

In our study, the designed parts were manufactured at constant 3D printing parameters, i.e., the sintering conditions were not changing. The influence of the parameters of the sintering process on the porosity was not observed. The shape of these components was the same as the shape used for the tensile test. Samples were manufactured with three different thicknesses (1, 2, 3 mm). The samples were printed in different directions with respect to the machine platform, specifically, in the XZ and Z directions. The CT measurement method was used to measure the porosity of components manufactured by the SLM additive technology. Porosity data were evaluated for individual part thicknesses and for individual part orientations in the machine. The main goal of the study was to determine how the orientation of the part during its production affects the porosity.

## 2. Materials and Methods

The 316L-0407 powder from Renishaw was used in this study. This is an austenitic stainless steel. The applications of this steel are in the plastic industry and die casting molds, dies for extrusion, instruments for surgery, and parts for the navy. The material composition is in Table 1 [19].

All parts were produced by using the Renishaw AM400 (Wotton-under-Edge, UK) machine. The parameters of the process used in this article were power of laser 200 W, speed of scanning 650 mm/s, and thickness of layer 50 µm. The investigated parts were printed with orientation in two directions—Z and XZ. Material thicknesses of printed parts were 1, 2, 3 mm. (Figure 1).

Different scanning strategies were used to produce components by the SLM method. By applying these strategies, we obtained a finer grain structure [20,21]. In this way, it is possible to produce components with thin walls [22]. Parts for our experiment were made using a “strip” strategy of scanning (Figure 2).

A non-destructive method for evaluation of the components’ porosity was used in this study. This method is suitable for measuring dimensions, but also for measuring the structures of materials. Goméz et al. compared measurements by CT techniques and the coordinate measuring machine (CMM). They also discussed standards for estimating measurement uncertainty [23]. The case studies [24,25] compared the CT method and measurement techniques with classical metrology implemented on CMM. They pointed out the advantages of using CT measurement methods. Measurement strategies using CT were investigated and evaluated. The result was evidence of the suitability of CT measurement for production processes, dimensional measurement, and structural control (external, internal).

Experimental investigation was performed on the device METROTOM 1500 from Zeiss (Oberkochen, Germany), using computed tomography. This device consists of these main parts: X-ray tube, rotational table, and detector, which is used for capturing two dimensional images. Software used for scanning and getting data was METROTOM OS 2.8.

The X-ray set ups were made according to producer recommendations and skills of the operator:Voltage: 180 kVCurrent: 900 µAResolution: 1024 × 1024 pxVoxel size: 119.43 µmNr of projections: 1050Integration time: 2000 ms

A cupper filter with thickness of 3 mm was used. The distance between the X-ray source and the scanning part was 450 mm.

Three dimensional models were evaluated in the VGStudio MAX 3.0 software (Volume Graphics, Heidelberg, Germany) after reconstruction. The first step was surface determination for recognition of the shape of the part. After that, 2 × compatibility porosity analysis was used. Its application shows pores scanned in each part. The principle of non-destructive measurement by CT is shown in (Figure 3). The position of the parts located in the computed tomography device is shown in (Figure 4).

## 3. Results

The first step was surface determination for recognition of the shape of the part. After that, 2 × compatibility porosity analysis was used. Its application shows pores scanned in each part. Figure 5 shows the pores in the part, which had a thickness of 1 mm and an orientation of layers in the Z direction.

Identification of defects at selected locations in the sample, which had a thickness of 1 mm and was oriented in the Z direction, is shown in Figure 6.

Figure 7 shows the statistics of evaluated values of material volume, defect volume, and defect volume ratio for a sample, which had a thickness of 1 mm and was Z oriented.

Figure 8 shows the pores in the 1 mm thick sample with an orientation of layers in the XZ direction.

The identification of the biggest defects at specific locations of the sample with 1 mm thickness, which was oriented in the XZ direction, is shown in Figure 9.

Figure 10 shows the measured values of material volume, defect volume, and defect volume ratio for a sample, which had 1 mm thickness and was oriented XZ.

Parts with 2 and 3 mm thickness with orientation in the Z and X directions were evaluated in an identical way. The identification of defects at specific locations of the sample with 2 mm thickness, which was oriented in the Z direction, is shown in Figure 11. Figure 12 shows samples printed in the XZ direction.

Figure 13 shows the measured values of material volume, defect volume, and defect volume ratio for the sample with thickness of 2 mm, which was oriented in the Z direction. The measured values for the sample oriented XZ are shown in Figure 14.

The identification of defects at specific locations of the sample with 3 mm thickness, which was oriented in the Z direction, is shown in Figure 15.

The identification of defects at specific locations of the sample with thickness of 3 mm, which was oriented in the XZ direction, is shown in Figure 16.

Figure 17 shows the measured values of material volume, defect volume, and defect volume ratio for a sample with thickness of 3 mm, which was oriented in Z. Figure 18 shows the measured values of material volume, defect volume, and defect volume ratio for the sample with thickness of 3 mm, which was oriented in XZ.

## 4. Discussion

Porosity is the most common defect observed in components made by SLM technology. The degree of the porosity can be influenced by the parameters of the laser process. Li et al. [26] described the influence of laser processing parameters on the porosity of additive manufactured parts; the material used was powder of 316L steel. The relationship between laser energy density and porosity was investigated. It was found that the porosity distribution is uniform in the layers. Simchi [27] investigated the effect of laser sintering parameters on the properties of different powders (Fe, Fe-C, Fe-Cu), and additionally, for 316L steel and M2 steel. As a result, it was determined that there is a relationship between density and laser power. The research by Kruth et al. [28] into stainless steel components (316L) demonstrated a relationship between scanning speed and grain size. The lower scanning speed causes the formation of larger grains and, thus, large defects. The research by Cherry et al. [9] found that total porosity is strongly influenced by the density of the laser energy. The porosity is high when the laser energy is low. As the energy density increases, the number of pores decreases. The smallest pore volume was observed for an energy of 104.52 J/mm^3^. Yosuf et al. [15] performed measurements where they applied the Archimedes method and the CT scanning method, which uses X-rays (XCT). The porosity of parts and their microhardness was investigated in this work. The parts were made of 316L steel by the SLM method. High density values (>99%) were found in the samples. A low porosity value was found (~0.82%). It was stated that this low pore volume did not affect the mechanical properties of the parts produced by the additive technology from 316L material. Al Faifi [29] identified the most statistically significant parameters of laser processing. The article identified the relationship between the number of pores and the parameters of the sintering process. It was found that the distance of the point, the time of exposure, and the thickness of the layer affect the parts density. Tolosa et al. [14] examined parts intended for mechanical testing. The parts were made by the additive SLM method. Different production methods were used with different orientations of the parts in space. It was found that the tensile properties of the parts and the strength of the steel parts produced by SLM are better than the properties of the steels produced by rolling. Using additive SLM technology, it is possible to produce complex shapes. Time savings and weight savings can be achieved. One article [13] dealt with the cracking of parts. This effect can be corrected by different part production strategies. This is a parameter of the laser scanning process. The production strategy affects the homogeneity of the structure. There is no need to change other process parameters. Tammas et al. [30] examined the distribution of defects in the volume of parts. They found that defect distribution is related to process parameters, contouring strategies, and section hatching. These parameters can affect the life of the parts. Hajnyš et al. [12] declared the most important parameters determining the strength of the material are scanning speed, then scanning strategy and ultimately laser power. The results of porosity have shown that the most important influences are scanning speed, laser power, and ultimately scanning strategy. In the article [31], the authors used CT method to examine porosity, pore sizes, and orientation. They used confocal microscopy to verify the CT results. The parts were made of 316L steel using SLM technology. Three samples were designed with different production orientations with respect to the machine platform. The smallest pores were detected in the sample oriented at an angle of 45° (0.15%). The highest porosity was detected in the sample oriented in YZ direction (2.97%). The porosity value for the sample oriented in the Z direction was 1.61%. These results are similar to the results from the experimental investigations presented in this paper. Additionally, higher porosity was achieved for the XZ-oriented component than for the Z-oriented component.

All the studies mentioned in the discussion point to how porosity is formed and how the porosity of components manufactured by additive technologies can be affected. There were also studies that report how porosity affects the material and its mechanical properties. Finally, the individual parameters of the laser process were identified here, such as scanning speed, power of laser, and strategy of scanning, which affect the porosity of the part and thus the properties of the part.

In this study, we pointed out the fact that, in addition to the parameters of the laser process, it is necessary to consider the layer orientation of the components in their production. Here, we examined the porosity of the samples and its relationship with respect to the layer orientation of the components.

In further research, we will submit the samples to other tests. We will investigate their properties by analyzing the microstructure and other material properties.

## 5. Conclusions

In this article, the porosity of components that were made using SLM technology was examined. The samples were produced with layer orientation in the Z and XZ directions with respect to the SLM machine platform. The experimental samples had thicknesses of 1, 2, and 3 mm.

The scan of the sample with thickness of 1 mm, which was printed in the Z direction shows that the pores were detected only on one side of the part. Pores of different sizes were identified, as shown in (Figure 6). The aim of this study was to compare the defect volume ratio for individual components oriented in different directions with respect to the machine platform. In this case, a defect volume ratio of 0.09% was found.

Other findings were shown in the 1 mm thick sample with orientation in the XZ direction. The pores were distributed throughout the entire volume of the part. The pore sizes were different, as shown in (Figure 9). The defect volume ratio for this sample was 0.99%.

Further measurements were performed on the sample with thickness of 2 mm oriented in the Z and XZ directions. Only a small number of pores were detected in the part that was oriented in the Z direction. These pores were located only on one side of the part, as it was in the case of the sample with thickness of 1 mm. Those pores are shown in (Figure 11). The defect volume ratio was 0.05%. Measurements for the part oriented in the XZ direction showed that the pores are distributed throughout the whole volume of the component (Figure 12). The defect volume ratio evaluated for this part was 0.62%.

The last experiments were performed on the samples with a thickness of 3 mm. The orientation of the parts in the machine during their production was in the Z direction and in the XZ direction. In the part oriented in the Z direction, essentially only one pore with a diameter of 0.46 mm was identified (Figure 15). The defect volume ratio was 0.00%. As it was with the previous samples oriented in the XZ direction, the pores in this part were distributed throughout the entire volume of the part. Again, pores of different sizes identified in this sample are shown in (Figure 16). The evaluated defect volume ratio was 0.54%.

Several factors contribute to the formation of porosity in the material. In addition to the parameters of the sintering process, in our case, using stripes as a scanning strategy also contributed. Its characterization is that the laser paths along the surface are divided into a cross-section. The scanning area is thus divided into small strips. These strips should overlap by default. This overlap may not be sufficient. This can cause the formation of pores. The area passed by the laser through the stripe strategy in the printing process of the part oriented in the Z direction is smaller than the area passed by the laser in the printing process of the part oriented in the XZ direction. The distribution of the area into strips is thus greater at XZ than at Z. This could be the cause of the lower porosity of the components produced in Z direction. Another possible cause of the porosity may be the scattering of the energy density. On a larger area, the scattering may not be sufficient, and the surface of the previous layer may melt. This does not create a coherent bond between adjacent surfaces, which can result in the formation of pores. Gases may be also the final factor for higher porosity detection. These gases could be either from the protective atmosphere or a product formed during the evaporation of the material. The assumption is that there is a greater chance of trapping gaseous particles within a larger scanned area. It can cause higher porosity. These stated assumptions need to be confirmed by further experiments.

Another fact we discovered that affects the porosity of the parts (it was not in the original plan and target) during this experimental investigation is the thickness of the sample. It is obvious from the measurements that with increased thickness of the sample, i.e., also with increased part volume, the number of pores in the part decreases. Probably, it is related to the sintering parameters. So far, it can only be expressed that, in the printing process of creating thicker samples, the laser beam acts longer on the surface. It can lead to increasing the melt temperature and improving the flow. The blanks are filled this way. Therefore, the proportion of the pores in the volume of material is reduced. This secondary finding can lead to enhanced experimental investigations about the relationship between the thickness of the part and the number of pores in the part.

It was found that the position of the part during its production affects its porosity. From the measured defect volume ratio values for individual sample thicknesses, it is possible to deduce that:Orientation of the sample during its production in the **Z** direction: as the sample thickness increased, the defect volume ratio in the sample volume decreased.sample thickness 1 mm (Defect volume ratio 0.09%),sample thickness 2 mm (Defect volume ratio 0.05%),sample thickness 3 mm (Defect volume ratio 0.00%).Orientation of the sample during its production in the **XZ**: as the sample thickness increased, the defect volume ratio in the sample volume decreased.sample thickness 1 mm (Defect volume ratio 0.99%),sample thickness 2 mm (Defect volume ratio 0.62%),sample thickness 3 mm (Defect volume ratio 0.54%).

Due to the defect volume ratio, the orientation of the sample in the Z direction during its production was more suitable than the orientation of the sample in the XZ direction during its production. This statement is valid in the relation to porosity evaluation. However, it may not be applied to other material properties.

## Figures and Tables

**Figure 1 materials-14-01142-f001:**
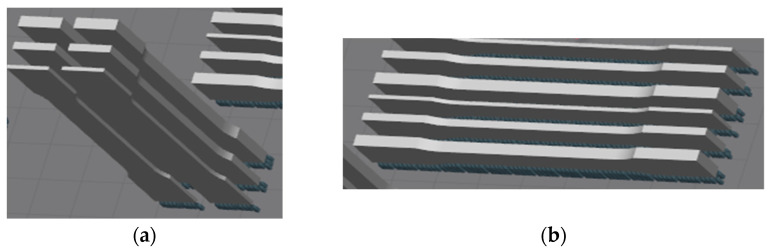
Investigated parts printed with different orientation with respect to the machine platform (**a**) Part orientation in Z direction; (**b**) Part orientation in XZ direction.

**Figure 2 materials-14-01142-f002:**
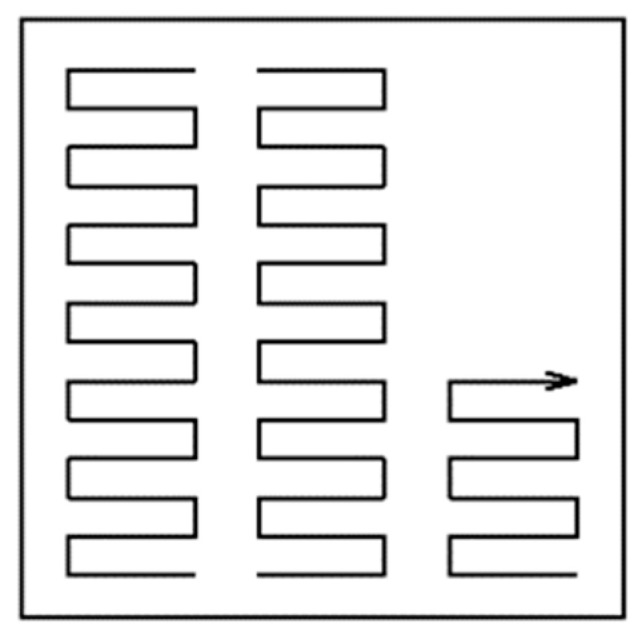
Strip scanning strategy used to build our components by the selective laser melting (SLM) method.

**Figure 3 materials-14-01142-f003:**
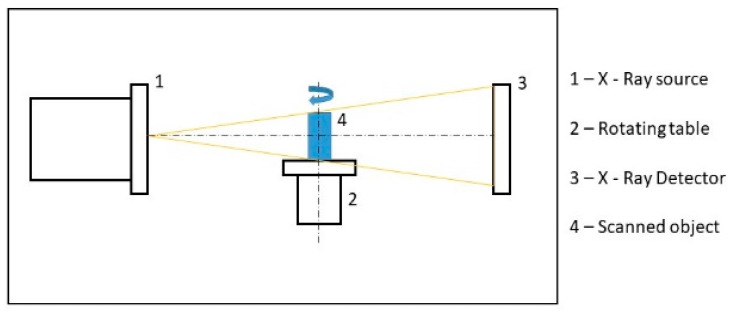
Principle of measurement by industrial computer tomography (CT).

**Figure 4 materials-14-01142-f004:**
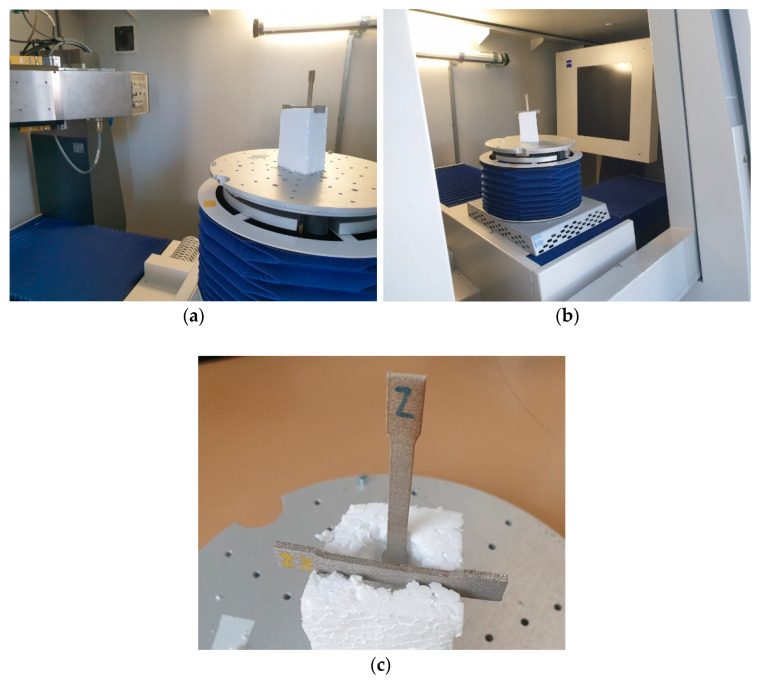
The position of the parts in the computer tomography device (**a**) View of the source of the X-rays, (**b**) View of the X-ray detector, (**c**) View of the experimental parts.

**Figure 5 materials-14-01142-f005:**
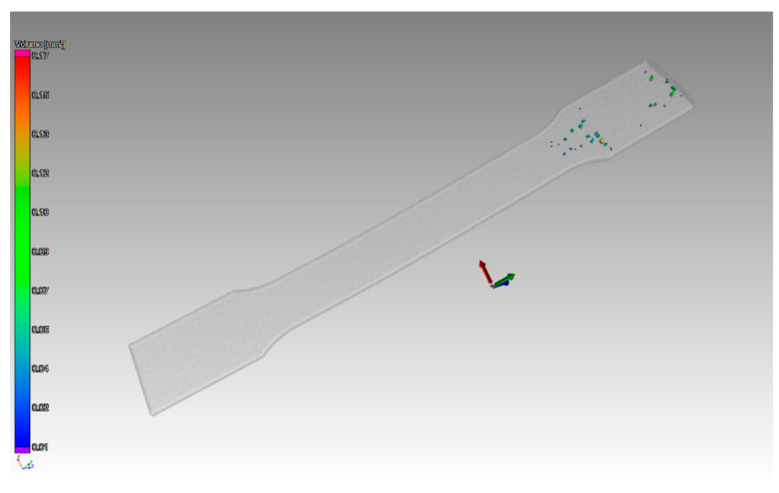
Pores detected in a part, which had a thickness of 1 mm oriented in Z.

**Figure 6 materials-14-01142-f006:**
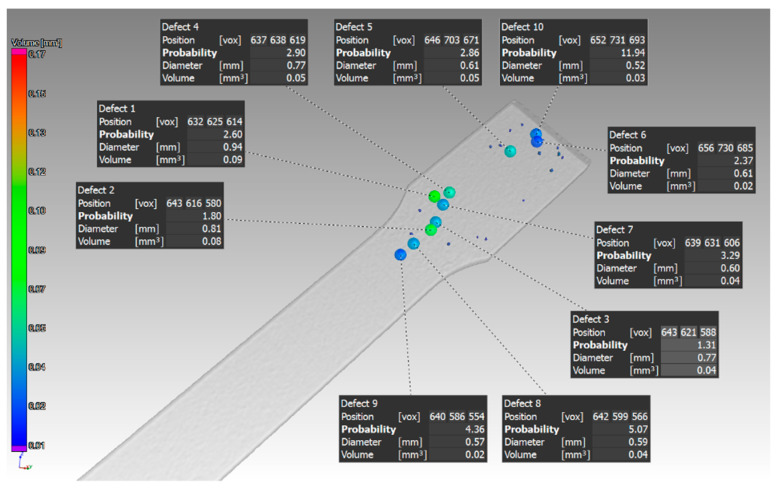
The biggest defects in the sample, which had a thickness of 1 mm and was oriented in Z.

**Figure 7 materials-14-01142-f007:**
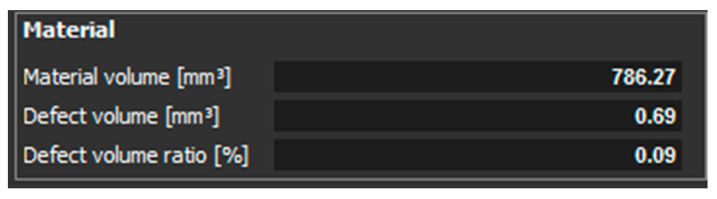
Evaluated values for a sample with a thickness of 1 mm, which is oriented in Z.

**Figure 8 materials-14-01142-f008:**
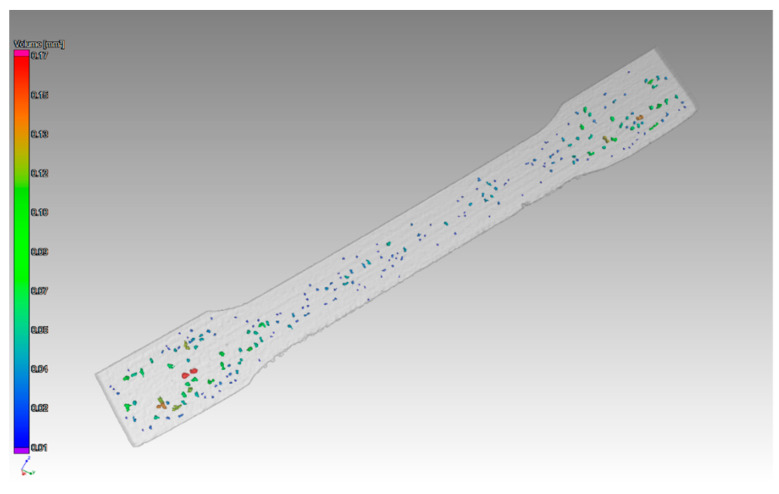
Defects detected in the sample with thickness of 1 mm printed in XZ directions.

**Figure 9 materials-14-01142-f009:**
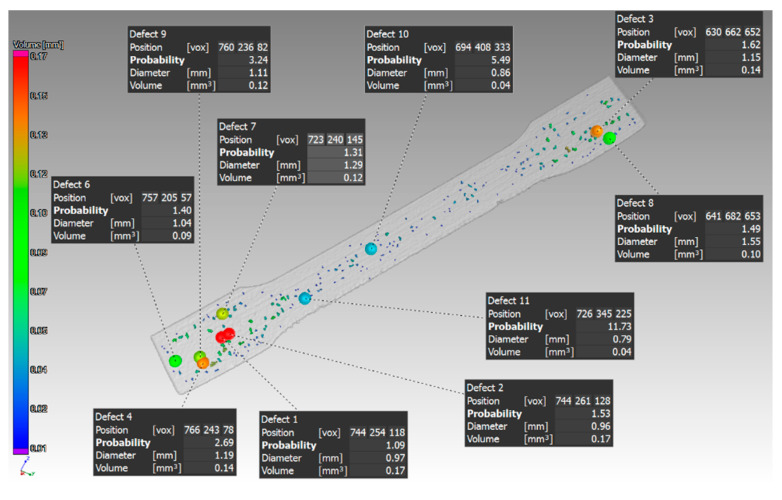
Specific defects evaluated in the sample, which had 1 mm thickness and was oriented XZ.

**Figure 10 materials-14-01142-f010:**
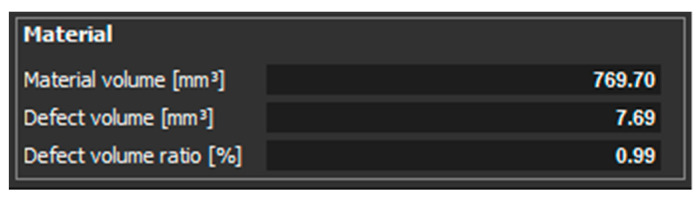
Measured values for a sample with a thickness of 1 mm and oriented XZ.

**Figure 11 materials-14-01142-f011:**
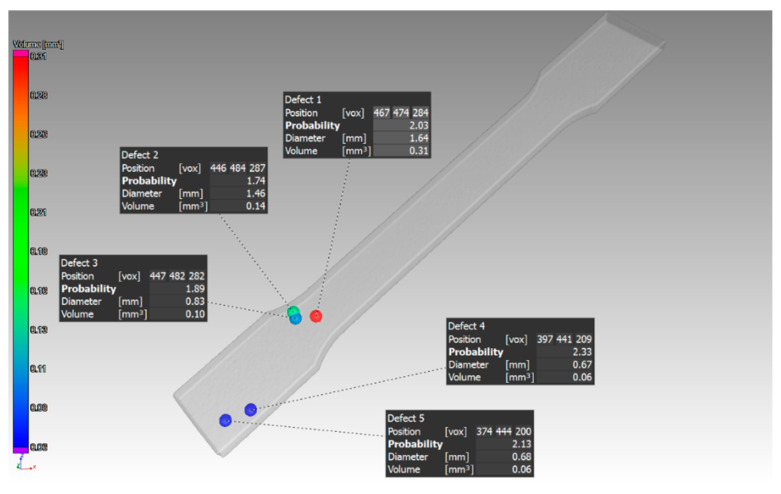
Specific defects on a sample with 2 mm thickness, which was oriented in the Z direction.

**Figure 12 materials-14-01142-f012:**
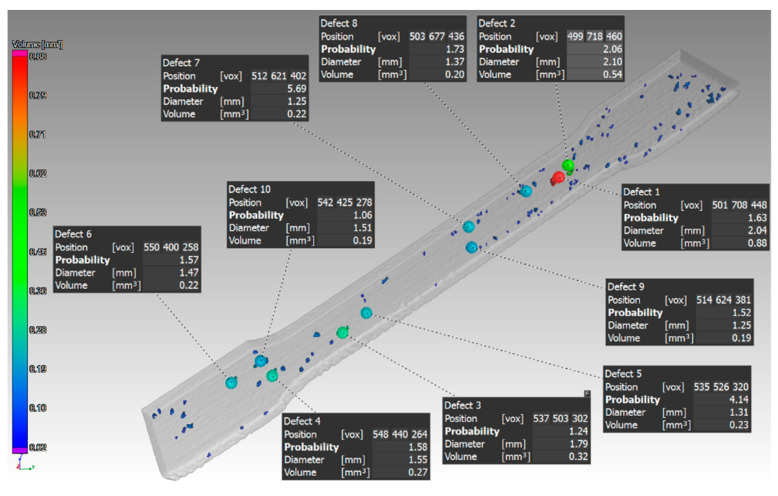
Specific defects on a sample with 2 mm thickness, which was oriented XZ.

**Figure 13 materials-14-01142-f013:**
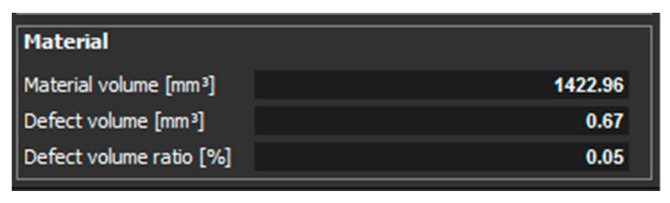
Measured values for a sample with 2 mm thickness, which is oriented in Z.

**Figure 14 materials-14-01142-f014:**
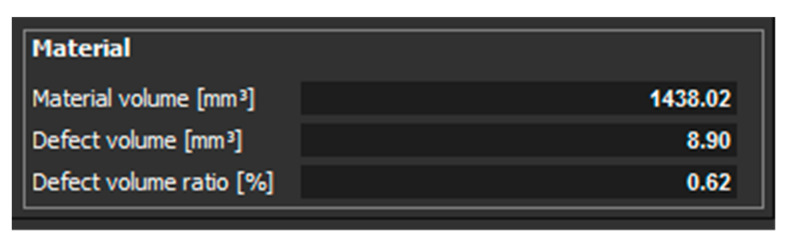
Measured values for a sample with 2 mm thickness, which is oriented in XZ.

**Figure 15 materials-14-01142-f015:**
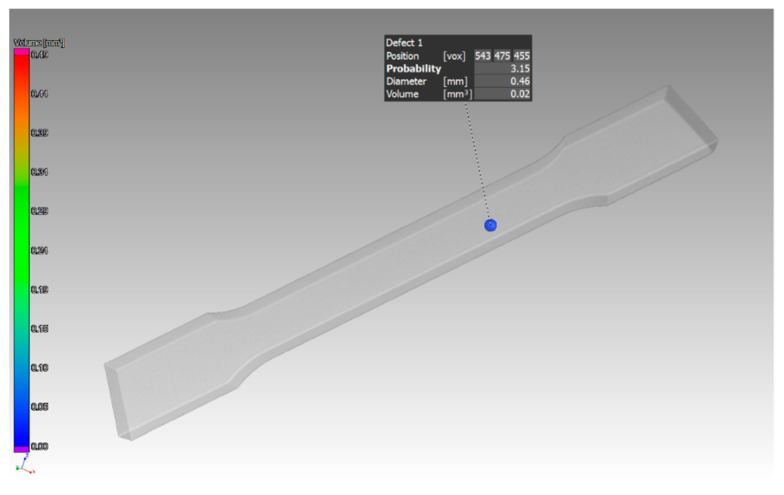
Specific defects in the sample with 3 mm thickness, which was oriented in Z.

**Figure 16 materials-14-01142-f016:**
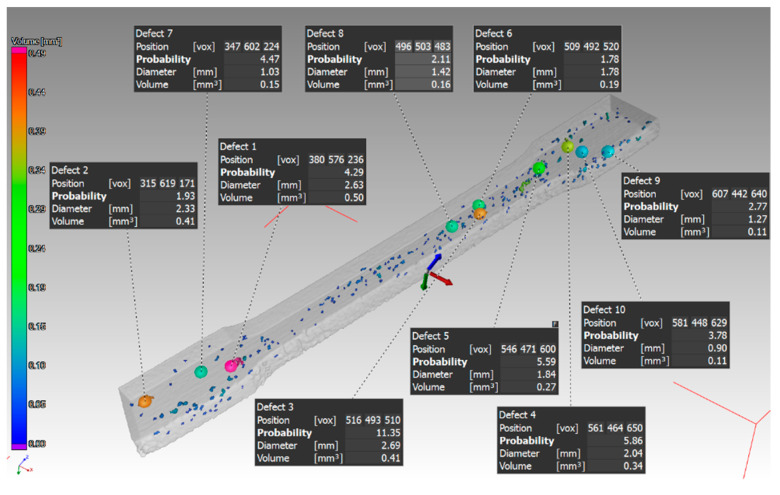
Specific defects in the sample with thickness of 3 mm, which was oriented in XZ.

**Figure 17 materials-14-01142-f017:**
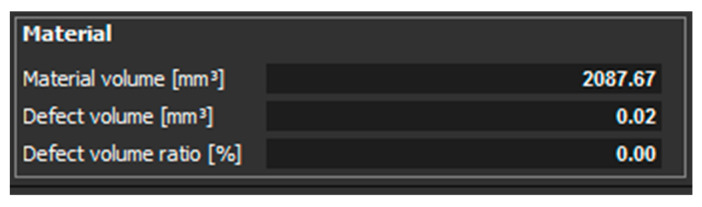
Measured values for the sample with thickness of 3 mm, which was oriented in Z.

**Figure 18 materials-14-01142-f018:**
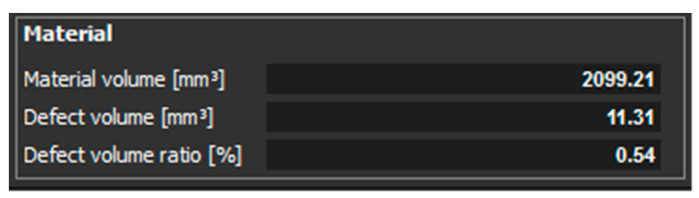
Measured values for the sample with thickness of 3 mm, which was oriented in XZ.

**Table 1 materials-14-01142-t001:** Chemical components of the 316L-0407 powder used for experiments.

Element	Fe	Cr	Ni	Mo	Mn	Si	P	C	S
Mass (%)	Balance	16–18	10–14	2–3	≤2	≤1	≤0.045	≤0.03	≤0.03

## Data Availability

Data available in a publicly accessible repository.

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
