# Peer review of "Porosity Analysis of Additive Manufactured Parts Using CAQ Technology"

_materials, 2021, doi:10.3390/ma14051142_

Round 1

Reviewer 1 Report

There are some weaknesses through the manuscript which need improvement. Therefore, the submitted manuscript cannot be accepted for publication in this form, but it has a chance of acceptance after a major revision. My comments and suggestions are as follows:

1- Abstract gives information on the main feature of the performed study, but some details about the specimens and experimental tests (at least in a couple of sentences) should be added. However, a concise abstract is needed.

2- Authors must introduce abbreviation (e.g., CAQ, CAM and SLM in introduction). It is not necessary to mention the name of companies in abstract and introduction.

3- Authors must clarify necessity of the performed research (in introduction). Aims and objectives of the study, must be clearly mentioned in concluding part of introduction.

4- The literature study must be enriched. It is highly recommended to read and cite the published papers: (a) https://doi.org/10.1016/j.compstruct.2020.112710 and (b) https://doi.org/10.1007/s10921-020-00721-1 Anyway, introduction is too short.

5- The presented table (Table 1) is a figure. It must be a real table. Figures must be illustrated in a proper size. For instance, Fig. 1 is too large.

6- It would be nice if authors illustrate a figure to show some technical details of the device (instead of Fig. 3).

7- Since it is an experimental investigation, it is necessary to add figures to show experimental conditions (e.g., printed specimens)

8- More details on experimental test must be presented. For instance, details of camera, taking photo, and recording videos.

9- In its language layer, the manuscript should be considered for English language editing. There are sentences which have to be rewritten.

10- The conclusion is too short and it must be more than just a summary of the manuscript. Please provide all changes in text and reference update (based on recommended papers) by red color in the revised version.

Author Response

Thank you very much for taking the time to review our article. Thank you very much for your comments and suggestions on how to edit the article. In the following text, we would like to offer you our reactions to your comments.

Abstract gives information on the main feature of the performed study, but some details about the specimens and experimental tests (at least in a couple of sentences) should be added. However, a concise abstract is needed.

Our reaction:

The abstract was supplemented with brief information about the shape of the part, printing parameters and radiation parameters.

Authors must introduce abbreviation (e.g., CAQ, CAM and SLM in introduction). It is not necessary to mention the name of companies in abstract and introduction.

Our reaction:

The designation CAQ has been added and explained in the abstract. The designation CAM and SLM are explained in the introduction text.

Authors must clarify necessity of the performed research (in introduction). Aims and objectives of the study, must be clearly mentioned in concluding part of introduction

Our reaction:

The final part of the introduction was supplemented by aims and objectives. There is also a motivation for this research.

The literature study must be enriched. It is highly recommended to read and cite the published papers: (a) https://doi.org/10.1016/j.compstruct.2020.112710 and (b) https://doi.org/10.1007/s10921-020-00721-1 Anyway, introduction is too short.

Our reaction:

The recommended literature has been added to the introduction of the article under numbers [17] and [18]. These resources have also been added to the references. The introduction was enriched with additional information from the literature.

The presented table (Table 1) is a figure. It must be a real table. Figures must be illustrated in a proper size. For instance, Fig. 1 is too large.

Our reaction:

Table 1 and figures have been modified.

It would be nice if authors illustrate a figure to show some technical details of the device (instead of Fig. 3).

Our reaction:

Figure 3 has been changed. Now shows the principle of measurement by industrial CT. Figure 4 has been added, showing the position of the parts in the computed tomography device.

Since it is an experimental investigation, it is necessary to add figures to show experimental conditions (e.g., printed specimens)

Our reaction:

Figure 4c showing experimental samples has been added and Figures 4a and 4b show important parts of CT (X-ray source and X-ray detector).

More details on experimental test must be presented. For instance, details of camera, taking photo, and recording videos.

Our reaction:

Figure 4 shows the details of the CT machine and the measured components. The text in the Materials and Methods section has been modified.

In its language layer, the manuscript should be considered for English language editing. There are sentences which have to be rewritten.

Our reaction:

The English was checked by two experts, for which we thank them very much. We hope it's okay.

The conclusion is too short and it must be more than just a summary of the manuscript. Please provide all changes in text and reference update (based on recommended papers) by red color in the revised version.

Our reaction:

The conclusions were enriched with a summary of the experimental investigation and the possible causes of lower porosity with components oriented in the Z direction. Also with possible causes of lower porosity in thicker parts.

Thank you very much again

Sincerely

team of authors

Reviewer 2 Report

In this manuscript, authors reported the analysis of CAQ technology results which is porosity of additive manufactured parts. The authors deal with porosity evaluation of metal components that produced by additive technologies using the Selective Laser Melting (SLM) method. However, I do not recommend this work to be accepted in current form. Here are the main reasons.

1. Their experiment is not correlated their claims. Authors wants to compare the porosity of their samples according to effects of particle size, particle shape, sintering temperature and sintering time. But there are no data to prove the correlation of them with porosity.

2. They introduced many conditions impacted on porosity but the authors only conducted the only one value test (thickness), which has not enough. Also, there are no enough explanation that why exposed layer orientation in the Z direction has lower defect volume ratio. Also, the author should descript the reasons why it has lower defect volume ratio according to increasing the thickness of sample.

3. There are 3 strategy in SLM method. The author using the “Stripes” scan strategy for the experiment. But there are any reason how the strategy reducing the porosity.

Author Response

Thank you very much for taking the time to review our article. Thank you very much for your comments and suggestions on how to edit the article. In the following text, we would like to offer you our reactions to your comments.

Their experiment is not correlated their claims. Authors wants to compare the porosity of their samples according to effects of particle size, particle shape, sintering temperature and sintering time. But there are no data to prove the correlation of them with porosity.

Our reaction:

The final part of the introduction was supplemented by aims and objectives. There is also a motivation for this research.

“In this paper, we focus on the research of porosity in components made by additive technology. Specifically, SLM technology. The motivation for this research was the study of materials and scientific publications on the porosity of materials produced by SLM. These studies have shown that porosity needs to be deeper examined given that it can significantly affect the mechanical properties of parts. The study of porosity and other properties of parts made by 3D printing must lead to quality products that will be produced in a shorter time.

In our study, the designed parts were manufactured at constant 3D printing parameters, i.e.: the sintering conditions were not changing. The influence of the parameters of the sintering process on the porosity was not observed. The shape of these components is the same as the shape used for the tensile test. Samples were manufactured with three different thicknesses (1, 2, 3 mm). The samples were printed in different directions with respect to the machine platform. Specifically, in the XZ and Z directions. The CT measurement method was used to measure the porosity of components manufactured by the SLM additive technology. Porosity data were evaluated for individual part thicknesses and for individual part orientations in the machine. The main goal of the study is to determine how the orientation of the part during its production affects the porosity.”

They introduced many conditions impacted on porosity but the authors only conducted the only one value test (thickness), which has not enough. Also, there are no enough explanation that why exposed layer orientation in the Z direction has lower defect volume ratio. Also, the author should descript the reasons why it has lower defect volume ratio according to increasing the thickness of sample.

Our reaction:

The main goal of the study is to determine how the orientation of the part during its production affects the porosity. The samples were oriented in different directions with respect to the SLM machine platform. Specifically, in the XZ and Z directions.

The conclusions were enriched with a summary of the experimental investigation and the possible causes of lower porosity with components oriented in the Z direction. Also with possible causes of lower porosity in thicker parts.

“Several factors contribute to the formation of porosity in the material. In addition to the parameters of the sintering process, in our case stripes as a scanning strategy. Its characterization is that the laser paths along the surface are divided into a cross section. The scanning area is thus divided into small strips. These strips should overlap by default. This overlap may not be sufficient. This can cause the formation of pores. The area passed by the laser through the stripe strategy in the printing process of the part oriented in the Z direction is smaller than the area passed by the laser in the printing process of the part oriented in the XZ direction. The distribution of the area into strips is thus greater at XZ than at Z. This could be the cause of the lower porosity of the components produced in Z direction. Another possible cause of the porosity may be the scattering of the energy density. On a larger area, the scattering may not be sufficient and the surface of the previous layer may melt. This does not create a coherent bond between adjacent surfaces, what can be resulting the formation of pores. Gases may be also the last factor for higher porosity detection. These gases could be either from the protective atmosphere or they can be product during the evaporation of the material. The assumption is that there is a greater chance of trapping gaseous particles with a larger scanned area. It can cause higher porosity. These stated assumptions need to be confirmed by further experiments.

It was discovered another fact, which affects the porosity of the parts (it was not in the original plan and target) during this experimental investigation. It is the thickness of the sample. It is obvious from the measurements that with increasing of thickness of the sample, i.e. also with increasing of the part volume, the number of pores in the part decreases. Probably, it is related to the sintering parameters. So far, it can only be expressed that in the printing process of the samples with bigger thickness, the laser beam acts longer on the surface. It can lead to increasing of the melt temperature and improving the flow. The blanks are filled this way. So the proportion of the pores in the volume of material is reduced. This secondary finding can lead to enhanced experimental investigations about the relation between the thickness of the part and the number of pores in the part.”

There are 3 strategy in SLM method. The author using the “Stripes” scan strategy for the experiment. But there are any reason how the strategy reducing the porosity.

Our reaction:

Figure 2 has been changed. Only one "stripes" strategy was used. The question of strategy reducing porosity has not been the subject of this experimental research. The main goal of the study is to determine how the orientation of the part during its production affects the porosity

Thank you very much again

Sincerely

team of authors

Round 2

Reviewer 1 Report

The paper has been improved and corresponding modifications have been conducted. In my opinion, the current version can be considered for publication in Materials.

Reviewer 2 Report

The manuscript is well revised. I recommend the manuscript for publication in Materials as it is.